# Phytochemical Profiling, Antioxidant, Anti-Inflammatory, Thrombolytic, Hemolytic Activity In Vitro and In Silico Potential of *Portulacaria afra* 

**DOI:** 10.3390/molecules27082377

**Published:** 2022-04-07

**Authors:** Sobia Tabassum, Saeed Ahmad, Kashif ur Rehman Khan, Fouzia Tabassum, Anjum Khursheed, Qamar uz Zaman, Najat A. Bukhari, Alanoud Alfagham, Ashraf A. Hatamleh, Yinglong Chen

**Affiliations:** 1Department of Pharmaceutical Chemistry, Faculty of Pharmacy, The Islamia University of Bahawalpur, Bahawalpur 63100, Pakistan; rsahmad_iub@yahoo.com (S.A.); anjumkhursheedrana@gmail.com (A.K.); 2Department of Botany, University of Agriculture Faisalabad, Faisalabad 38000, Pakistan; fouzia8447@gmail.com; 3Department of Environmental Sciences, The University of Lahore, Lahore 54590, Pakistan; qamar.zaman1@envs.uol.edu.pk; 4Department of Botany and Microbiology, College of Science, King Saud University, P.O. Box 2455, Riyadh 11451, Saudi Arabia; najatab@ksu.edu.sa (N.A.B.); aalfaghom@ksu.edu.sa (A.A.); ahatamleh@ksu.edu.sa (A.A.H.); 5The UWA Institute of Agriculture, School of Agriculture and Environment, The University of Western Australia, Perth, WA 6009, Australia; yinglong.chen@uwa.edu.au

**Keywords:** *Portulacaria afra*, phytochemicals, docking techniques, hemolytic activity, anti-inflammatory, GC-MS

## Abstract

The use of complementary herbal medicines has recently increased in an attempt to find effective alternative therapies that reduce the adverse effects of chemical drugs. *Portulacaria afra* is a rich source of phytochemicals with high antioxidant activity, and thus may possess health benefits. This study used the latest developments in GC-MS coupling with molecular docking techniques to identify and quantify the phytoconstituents in *P. afra* tissue extracts. The results revealed that *n*-butanol *P. afra* (BUT-PA) dry extracts contained total phenolic and flavonoids contents of 21.69 ± 0.28 mgGAE/g and 196.58 ± 6.29 mgGAE/g, respectively. The significant potential of antioxidants was observed through CUPRIC, FRAP, and ABTS methods while the DPPH method showed a moderate antioxidants potential for *P. afra*. Enzymatic antioxidants, superoxide dismutase, peroxidase and catalase also showed a better response in the BUT-PA dry extracts. The thrombolytic activity of the BUT-PA extracts ranged from 0.4 ± 0.32 to 11.2 ± 0.05%. Similarly, hemolytic activity ranged from 5.76 ± 0.15 to 9.26 ± 0.15% using the standard (triton x) method. The BUTPA and CHPA showed moderate acetylcholinesterase and butrylcholinesterase inhibition, ranging from 40.78 ± 0.52 to 58.97 ± 0.33, compared to galantamine. The carrageenan induced hind-paw edema assay, while BUT-PA extracts showed anti-inflammatory properties in a dose-dependent manner. Furthermore, 20 compounds were identified in the BUTPA extracts by GC-MS. Molecular docking was performed to explore the synergistic effect of the GC-MS-identified compounds on COX-1 and COX-2 inhibition. A high binding affinity was observed for Stigmastan-3, 5-diene, Phthalic acid, 3. Alpha-Hydroxy-5, 16-androstenol. The computed binding energies of the compounds revealed that all the compounds have a synergistic effect, preventing inflammation. It was concluded that active phytochemicals were present in *P. afra,* with the potential for multiple pharmacological applications as a latent source of pharmaceutically important compounds. This should be further explored to isolate secondary metabolites that can be employed in the treatment of different diseases.

## 1. Introduction

Medicinal plants are a rich source of novel lead compounds that can be used for a variety of therapeutic and pharmacological purposes [1]. The World Health Organization (WHO) estimates that up to 80% of the population in developing countries still depend on local medicinal plants to fulfill their primary healthcare needs. An estimated 25% of prescription drugs and 11% of drugs considered essential by the WHO are derived from plants, and a high number of synthetic drugs are obtained from precursor compounds originating from plants [2]. Their easy accessibility, efficacy in treatment and affordable cost are the main reasons to prefer traditional medicine to modern medication [3]. Thousands of polyphenolics compounds, such as phenolic and flavonoids, have been identified in these medicinal plants to date. The pharmacological and physiological potentials of phenolic compounds rely on their free radical scavenging and antioxidant activities to maintain the activity of the enzymes responsible for detoxification. Inflammation is a ubiquitous progression that occurs in a disturbed state of homeostasis, such as damage exposure to contaminating substances [4] and infection [5]. It is often triggered by innate immune system receptors to remove pathogens [6] when they are characterized by pain [7], redness, heat or warmth, and swelling [8]. Inflammation is classified into two types: acute and chronic. Acute inflammation could be the body’s early response to potentially damaging stimuli [9]. The symptoms of inflammation are severe pain, swelling, and loss of function of the affected area [10].

The inflammatory response is out of proportion in chronic inflammation, resulting in bodily harm [11]. Cyclooxygenase (COX) is a major enzyme in the production of prostacyclins, prostaglandins, and thromboxane, which play a role in inflammation, pain, and platelet aggregation [12,13]. Enzyme-glucuronidase has also been identified as a mediator in the beginning and during the progression of inflammation [14]. Free radicals include reactive species such as hydroxyl radicals, peroxyl radicals, superoxide radicals, hydrogen peroxide, superoxide anion, and other lipid peroxides [15].

*Portulacaria afra* is one of the dominant medicinally important plants, widely grown in the east of South Africa, belonging to the family Didiereaceae [16]. In this moist climate, it is relatively rare, and it tends to favor dryer rocky outcrops and slopes [17]. *P. afra* is an annual succulent herb with thick fleshy leaves for water storage [18]. This plant can grow up to 2 meters tall. It has red stems with round fleshy leaves, and produces pink flowers. It widely grows in warm climates in eastern South Africa [19]. The genus *Portulaca* contains approximately 100 species with a wide distribution, primarily in tropical and subtropical regions, and has a high degree of morphological variability [20].

Traditionally *P. afra,* was used to treat various skin disorders sores, ringworm, boils, burns, rash, wounds shingles, warts abscesses, and acne [21,22]. The identification of active phytochemicals from these natural sources, which can block COX-2 activity, may aid the production of anti-inflammatory pharmaceuticals. The present study aimed (1) to evaluate phytochemical components, total phenolic, total flavonoids, non-enzymatic antioxidants (DPPH, FRAP, ABTS, CUPRIC), enzymatic antioxidants (SOD, POD, CAT) and pharmacological activities in various fractions (*n*. hexane chloroform, *n*. butanol, methanol) using gas chromatography mass spectrometry (GC-MS), and (2) to perform in silico molecular docking analysis of bioactive compounds identified from *P. afra.*

## 2. Results

### 2.1. Phytochemical Screening, Total Phenolic and Flavonoids Contents

The presence of glycosides, saponins, tannins, flavonoids, phenols, fixed oils, and carbohydrates was discovered during phytochemical screening (Table 1). The total phenolic level identified in BUT-PA extracts of *P. afra* was 21.69 ± 0.28 mgGAE/g and flavonoid content was 196.58 ± 6.29 mgGAE/g in the dry extracts.

### 2.2. Antioxidant Potential 

Analysis showed that the BUT-PA extracts of *P. afra* had significant antioxidant potential though CUPRIC, FRAP and ABTS methods, and moderate antioxidant potential through the DPPH method (Figure 1). The BUT-PA extracts showed the maximum antioxidant potential 144.74 ± 2.20 through the FRAP method, followed by a moderate potential via CUPRAC 124.19 ± 5.34 and ABTS 126.22 ± 0.64 methods, respectively. Enzymatic antioxidant in the BUT-PA extracts of *P. afra* had good potential for all tested enzymes: catalase (CAT), peroxidase (POD), and superoxide dismutase (SOD) (Figure 1). 

### 2.3. Acetylcholinesterase (AChE) and Butrylcholinesterase (BChE) Inhibition Activity

The decreasing order for AChE inhibition of the extract/fractions were obtained as: *n*-butanol fraction (BUTPA) > chloroform fraction (CHPA) > *n*-hexane fraction (NHPPA) > methanolic extract (MPA) (Figure 2). The % age acetylcholinesterase inhibition activity of the extract/fractions of *P. afra* was in the range from 20.55 ± 0.5 to 58.97 ± 0.33. The BChE inhibition results of different fractions were ordered as follows: chloroform fraction (CHPA) > *n*-butanol fraction (BUTPA) > *n*-hexane fraction (NHPA) > methanolic extract (MPA). The % inhibition of BChE of *P. afra* ranged from 22.29 ± 1.927 to 57.64 ± 1.192 (Figure 2).

### 2.4. Thrombolytic Activity 

Thrombolytic activity was calculated for HPA, CHPA, MPA, and BUTPA. Thrombosis inhibition compared by Streptokinase (standard) showed significant clot lysis activity. While water was used as negative control, it showed minimum clot lysis. MPA showed minimum lysis 3.41 ± 0.32, followed by CHPA 7.32 ± 0.15 and HPA 11.21 ± 0.05. BUTPA showed a clot lysis of 5.51 ± 0.30, compared to the standard, 91.41 ± 0.06 (Table 2).

### 2.5. Hemolytic Activity

The data represented in Table 3 present the hemolytic activity of different extracts of *P. afra.* The MPA fraction has highest hemolytic percentage, 9.26 ± 0.15%, followed by BUTPA, 8.53 ± 0.25%, and CHPA, 7.46 ± 0.15%, which are statistically similar to each other. The lowest value, 5.76 ± 0.15%, was noticed in HPA extract. Overall, all four fractions have less than 30% hemolysis activity, so all fractions are nontoxic to humans and safe (Table 3).

### 2.6. GC–MS Profiling

The existence of plant metabolites with documented pharmacological potential was revealed by GC–MS analysis (Figure 3 and Table 4). Twenty compounds were identified from the NIST library, i.e., 1-Tetradecene (8.93%), Pentadecane (10.2%), 2,5-Di-tert-butylphenol (10.414%), 7-Hexadecene (11.293%), Hexadecane (11.374%), Heptadecane (12.485%), Phthalimide-acetonyl (12.95%), E-15-Heptadecenal (13.50%), 1,2-Benzenedicarboxylic acid (15.60%), Dichloroacetic acid (15.89%), Hexadecanoic acid, butyl ester (18.49%), Phenol,2,2’-methylenebis (21.90%), 9-O-Pivaloyl-N-acetylcolchinol (23.43%), Phthalic acid (23.68%), Stigmastan-3,5-diene (31.35%), alpha.-Hydroxy-5, 16-androstad (32.96%). The size of the percent area was observed as follows: compounds 3. Alpha.-Hydroxy-5, 16-androstad, (19.95%), Phthalic acid (4.61%), Stigmastan-3, 5-diene (3.92%), 9-O-Pivaloyl-N-acetylcolchinol (3.07%), respectively. The structure of the major compounds detected is given in Figure 3 and Figure 4 and Table 4.

### 2.7. Carrageenan-Induced Hind Paw Edema

The BUTPA extract showed anti-inflammatory effect at various doses, i.e., 25, 50, 100 mg/kg. At 25 mg/kg, BUTPA extracts showed anti-inflammatory activity after the 2nd, 3rd and 4th hour, i.e., 3.44 ± 0.25, 3.37 ± 0.31 and 3.15 ± 0.42, respectively (Figure 5). At 50 mg/kg, BUTPA extracts showed anti-inflammatory activity after the 1st, 2nd, 3rd and 4th hour, i.e., 3.69 ± 0.15, 3.25 ± 0.08, 3.96 ± 0.86 and 2.81 ± 0.136, respectively. At 100 mg/kg, BUTPA extracts showed anti-inflammatory activity greater than the standard after the 1st, 2nd, 3rd and 4th hour, i.e., 4.14 ± 0.52, 4.07 ± 0.45, 3.09 ± 0.23 and 3.7 ± 0.74, respectively (Figure 5).

### 2.8. Molecular Docking

To obtain a better insight into the inhibition ability of the studied compounds and to correlate the experimental enzyme inhibition results, three compounds from the GC-MS profile of BUTPA extract were docked against COX-1 and COX-2. Stigmastan-3, 5-diene showed the maximum binding affinity, i.e., −8.2 against COX-1, while it showed a −7.32 binding affinity against COX-2. Androstenol showed a −8.1 binding affinity against COX-1, while the maximum binding was shown against COX-2, i.e., (−9.1). Phthalimide, N-acetonyl showed a binding affinity of −8.1 against COX-1 and a binding affinity −8.2 against COX-1, and showed Van der wall forces and hydrogen bonding with amino acid (Table 5 and Figure 6).

## 3. Discussion

Antioxidants are substances that prevent or delay the oxidation progressions caused by atmospheric oxygen. They are crucial in an organism’s defense mechanism against pathologies caused by free radicals [35]. Therefore, for the determination of phytochemicals and antioxidant potential, BUT-PA extract of *P. afra* was prepared in methanol by maceration. Initially, the phytochemical investigation was carried out using the guidelines mentioned in the phytochemistry manual [36]. We obtained positive results for the secondary metabolites of tannin, flavonoids, alkaloids, Phytosterol, saponins, cardiac glycoside quinolones, and triterpenoids (Table 1). Our results are consistent with studies on Purslane [37]. Major constituents, such as alkaloids, contribute to analgesic and antimicrobial activity; flavonoids and tannins act as antioxidant and antibacterial agents and saponins have antibacterial, anti-inflammatory, anticancer, and anti-diabetic activities [38]. The presence of these phytochemicals in the extracts/fractions of *P. afra* might contribute to its therapeutic potential. Flavonoid and phenolic are active metabolites that are produced in response to environmental stress and have received a lot of attention from researchers due to their potential benefits [39]. In the current situation, total phenolic levels were observed in BUT-PA extract of *P. afra*, (22.69 ± 0.28 mgGAE/g) and flavonoids contents (197.58 ± 6.29 mgGAE/g) in dry extract. These higher values may be attributed to the higher polyphenols content in roots compared to leaves.

In this work, for the first time, BUT-PA extracts of *P. afra* were investigated for their antioxidant potency in terms of the potential non-enzymatic and enzymatic antioxidants to support its traditional medicinal usage. The BUT-PA extracts of *P. afra* were identified to have significant antioxidant potential though CUPRIC, FRAP and ABTS methods, and moderate antioxidant potential through the DPPH method, as in Figure 1. The results for antioxidant enzymes, BUT-PA extracts of enzymatic antioxidants of *P. afra* namely, catalase (CAT), peroxidase (POD), superoxide dismutase (SOD), showed good enzyme activities for all the tested enzymes. Our results agree with the previous finding, in which antioxidant activity was determined using different methods [40].

The most common type of dementia is Alzheimer’s disease (AD). In this regard, numerous synthetic and plant-derived cholinesterase inhibitors are ordinarily used for the management and betterment of the disease [41]. For the first time NHPA, CHPA, MPA and BUTPA, of *P. afra* were evaluated for their acetylcholinesterase and butrylcholinesterase activity. The results were expressed as the % inhibition of enzymes ± standard deviation Figure 2. CHPA and BUTPA showed a higher % of inhibition of both enzymes then NHPA and MPA when compared to their respective standard, i.e., the inhibition of Acetylcholinesterase and butrylcholinesterase was assessed against galantamine (standard). The BUTPA showed an inhibition of acetylcholinesterase and butrylcholinesterase ranging from (57.64 ± 1.192) to (58.97 ± 0.33), compared to the standard, which ranged from (98.14 ± 1.00) to (74.14 ± 1.18). The lowest % of inhibition of acetylcholinesterase and butrylcholinesterase was shown by HPA, i.e., (20.55 ± 0.5) to (22.29 ± 1.927), when compared to standard Galanatmine. The present investigation revealed that BUTPA successfully contributed the maximum potential, indicating the positive relationship between phenolic content and cholinesterase’s enzymes inhibition assays [42].

To the best of our knowledge, the literature does not report on the %age thrombolytic Activity of the methanolic extract, *n*-hexane, chloroform, and *n*- butanol fractions of *P. afra* from Pakistan. In the present study, the thrombolytic percentages of four different plant extracts, HPA, CHPA, MPA, BUT PA, are presented in Figure 3, with a comparison to Streptokinase, the positive control. From the results, it is apparent that Streptokinase has (91.36 ± 0.06) clot lysis. Meanwhile, HPA exhibited the highest activity (11.2 ± 0.05%), followed by CHPA (7.3 ± 0.15%) and BUTPA (5.5 ± 0.30%), whereas MPA presented the lowest activity, which was negligible (3.4 ± 0.32%). Therefore, BUTPA has no significant thrombolytic activity. Staphylokinase and streptokinase are microbial plasminogen activators, which act as co-factoring molecules to promote development [43]. Our results are in line with the previous data [44].

Toxicology tests can identify several of the problems that may result from the use of medicinal plants/herbs, particularly in vulnerable people [45]. Hemolysis is the rupturing of red blood cells (erythrocytes), which indicates the cytotoxic effects on red blood cells [46]. If the degree of hemolysis is greater than 30%, the plant extracts are deemed hazardous towards erythrocytes [34]. Table 2 presents the hemolytic activity of different extracts of *P. afra.* The MPA fraction has highest hemolytic percentage (9.26 ± 0.15%), followed by BUTPA (8.53 ± 0.25%), AND CHPA has (7.46 ± 0.15%), and HPA has the lowest hemolytic activity (5.76 ±0.15%), as shown in Table 3. Overall, all four fractions have less than 30% hemolysis activity, so all fractions are nontoxic to humans and safe. 

The genus Portulaca was shown to have anti-inflammatory properties by suppressing pro-inflammatory cytokine levels [47]. Certain natural fatty acid esters, such as methyl palmitate, have an inhibitory capacity against NF-B, which is accompanied by the downregulation of inflammatory channels [48]. The anti-inflammatory activity was measured in albino rats by measuring paw edema for four hours and inhibition was compared with standard indomethacin from 1- to 4-h intervals. The results exhibited that BUTPA extract at various doses (25, 50, and 100 mg/kg) had anti-inflammatory activity. The in vivo inhibition was assessed after 1 h, 2 h, 3 h, and 4 h. At 25 mg/kg, the BUTPA extract unveiled an anti-inflammatory effect. The BUTPA extract exhibited greater anti-inflammatory activity than the standard. At 400 mg/kg, the BUTPA extract showed an anti-inflammatory effect after the 2nd, 3rd, and 4th hour, as shown in Figure 5. Our results are consistent with the previous results for the same genus, which were reported by Reference [43].

The BUT-PA fraction of *P. afra* was subjected to gas chromatography-mass spectrometry (GCMS). The NIST library was used to identify the metabolites and 20 compounds were tentatively identified. The retention time in minutes (RT), area percentage (area %), compound name, molecular formula (M.F.), chemical class (Class), and molecular weight (M.W.) are shown in Table 3.The major identified compounds were Benzene, 1,3-dimethyl-, hexa- chloro derivative, *p*-Xylene, 2,5-Di-tert-butylphenol, Hexadecanoic acid, butyl ester, 9-O-Pivaloyl-N-acetylcolchinol, Phthalic acid, Stigmastan-3,5-diene, 3.alpha.-Hydroxy-5, Phthalimide, N-acetonyl, E-15-Heptadecenal, 1,2-Benzenedicarboxylic acid (Table 4 and Figure 3). 2,5-Di-tert-butylphenol showed antibacterial and antioxidant activity were as reported by Reference [49]. Aromatic compounds with methoxide group exhibit high antioxidant activity [50]. Such compounds have been identified in this study. Phenolic acids and their derivatives scavenge reactive oxygen species, showing antioxidant behavior. Compounds with one, two or three free hydroxyl groups on their aromatic ring exhibit low, medium and high antioxidant activity, respectively [51]. The highest antioxidant activity was observed in compounds with an -OH-group-oriented ortho or para position to the COOH group. The antioxidant activity decreases if the hydroxyl group is blocked, whereas no pronounced effect was served in the case of blocked carboxylic acid group regarding the compounds. Various environmental factors generate oxidative stress in plants. Consequently, the formation of phenolic metabolites with anti-oxidant properties is induced to overcome this stress.

The computational approach to molecular docking is now commonly employed in the drug development process. Docking has the advantage of determining the binding affinity of protein binding complexes, in this case, the chemical components from GCMS, as well as identifying the kind of interaction between the research molecules at the enzyme or receptor area through specific important interactions [52]. Molecular docking studies were performed for Cyclooxygenase-1 (COX-1) and cyclooxygenase 2 (COX-2). The PyRx virtual screening tool conducts flexible multiple ligand-docking studies within protein-binding sites [46]. A total of twenty compounds were docked against Cyclooxygenase 1 (PDB: 4O1Z) and Cyclooxygenase 2 (PDB: 5IKV). The hydrogen bond plays an essential role in protein–ligand interactions and other hydrophobic interactions, such as alkyl and pi alkyl, as well as ensuring the stable binding of ligands with proteins [53]. The best RMSD values were found with meloxicam and flurbiprofen for the corresponding protein targets: COX-1 and COX-2. We identified three compounds with the best energy affinity against these two proteins. With Cox-1 Stigmastan-3, 5-diene has the lowest binding affinity (−8.2), and alkyl and pi sigma interact with the amino acids. It is worth noting that the Stigmastan-3, 5-diene also had a lot of interactions with amino acids near to the COX-1 active site’s amino acids. As a result, this molecule could change the local structure, which might have a biological effect. We observed a hydrophobic alkyl-type interaction with distance 4.99A for the amino acid LEU99, 4.21A for the amino acid LEU92, 4.75 and 5.41A for the amino acid VAL116, 4.60A for the amino acid VAL119, 4.28 and 4.78A for the amino acid ILE89, and 3.9A for the amino acid TRP100. There was no hydrogen bonding in this. For the other two compounds, Androstenol and Phthalimide, N-acetonyl- had 8.1 binding affinities, while androstenol has one hydrogen bond with GLN44, with a distance of 2.37A°, and Phthalimide, N-acetonyl- has hydrophobic interactions with different amino acids. The Androstenol molecule had hydrophobic alkyl-like interactions with the amino acids CYS36, CYS47, and PRO153, with varying distances in the COX-2 protein. At a distance of 1.64A, we observed an unfavorable doner–doner type interaction for the amino acids ARG44. Another compound, such as Stigmastan-3,5-diene, has alkyl type interactions with 7.3 binding affinities, and Phthalimide, N-acetonyl- has pi alkyl and pipi t-shaped interactions with −8.2 binding affinities (Table 5 and Figure 6).

## 4. Materials and Methods

### 4.1. Collection of Plant Materials

Plant materials of *Portulacaria afra* were amassed in March, 2019, from the Baghdad Campus, The Islamia University of Bahawalpur, Pakistan. The plant was identified, and a specified voucher number of 314 was allotted by Taxonomist from the Life Sciences Department, The Islamia University of Bahawalpur, Pakistan. Manually separated leaves and shoots were rinsed with tap water, chopped into small pieces with a razor blade, and dried in fresh air under shade. The dried samples were mechanically ground with a grinder and sieved through a fine sieve (400 mesh numbers) to obtain a uniform particle size sample, which was then stored in airtight, black-coated glass container for further analysis.

### 4.2. Extraction Method 

Dried powder of the whole plant, of about 1 kg, was soaked in methanol for 72 h. This was routinely agitated through maceration. Afterward, the material was passed through muslin cloth to separate the material from solvent, and then the solvent was filtered with filter paper using a Buckner funnel. The obtained marc was then pre-soaked in *n*-butanol solvent for 72 h to collect all the remaining constituents from the plant material. This procedure was repeated for a third time to collect all the remaining constituents. The filtrate extracted with *n*-hexane, chloroform, and *n*-butanol solvent was concentrated using a rotary evaporator at 35 °C under lower pressure. The dried extract was weighed and stored in an airtight container for further analysis [54]. 

### 4.3. Chemicals and Drugs 

Indomethacin was obtained from Shanxi Guangsheng pharmaceutical, carrageenan (Sigma–Aldrich, St. Louis, MI, USA). Indomethacin (15 mg/kg) was prepared for delivery in double-distilled water.

### 4.4. Phytochemical Screening

Preliminary phytochemical analysis of *n*. butanol extract (BUTPA), methanol fraction (MPA), chloroform fraction (CHPA), *n*-hexane fraction of *P. afra* was performed by using previously established protocols described by [47]

### 4.5. Estimation of Total Phenolic Contents (TPC)

The total phenolic contents (TPC) from each of the aforementioned extracts and fractions were determined using a colorimetric Folin–Ciocalteu reagent (FCR) technique with modifications. Then, they were incubated for 10 min with 10 µL of diluted FCR (10%), 100 µL of sample solution, and 90 µL of 15% *w*/*v* aqueous sodium carbonate solution. This mixture was incubated for another 90 min at 37 °C. At 750 nm, absorbance was measured. They were both positive and negative controls (gallic acid). TPC was computed using a calibration curve (range 0–100 g) and reported as milligram gallic acid equivalents per gram of dry material. The tests were performed in duplicate as described by [43].

### 4.6. Estimation of Total Flavonoid Contents (TFC)

An aluminum chloride colorimetric test was used to assess the total flavonoid content (TFC), and the number of total flavonoids was reported in milligram of rutin equivalents (mgRE/g extract) [55,56]. A calibration curve with a range of from 0 to 100 l (0–100 g) was created, using rutin as the standard, at 1 mg/mL in methanol. All the solutions were made with methanol. A total of 100 µL of test solution was mixed with 25 µL of 1% sodium nitrite solution and left to stand for 5 min, before adding 10 µL of 1% aluminum chloride solution and leaving it to react for another 5 min. Finally, 35 µL of 4% was added. At 510 nm, absorbance was read. The calibration curve equation was used to calculate TFC values, which were represented as milligram rutin equivalent per gram of dry extract (mg of RE/g extract).

### 4.7. Determination of Antioxidant Potential 

The antioxidant activity of HMPA extract was assessed using two radical scavenging, i.e., DPPH, ABTS while reducing power was determined using, i.e., the protocol reported for FRAP and CUPRAC [57]. 

#### 4.7.1. DPPH Assay

For the DPPH assay radical scavenging assay, 100 µL of extract/fractions solution and 400 µL of DPPH were mixed in a 96-microtiter plate. This mixture was incubated at ambient temperature for 30 min in darkness. Absorbance was measured at a wavelength of 517 nm, with the help of a BioTek Synergy HT (Winooski, VT, USA) microplate reader. Results were expressed as Trolox equivalents per gram of extract (mg TE/g extract).

#### 4.7.2. ABTS Assay

For the ABTS reducing power assay, the formation of ABTS+ radical cation was due to the incubation of a mixture of 7 mM ABTS with 2.45 mM potassium persulfate in darkness at room temperature. The prepared stock solution of extract/fractions was diluted until its absorbance reached 0.700 ± 0.02 at 734 nm. A volume of 100 µL of extract/fraction solutions was combined with the previously prepared 200 µL ABTS+ solution in a 96-microtiter plate. This mixture was incubated at ambient temperature for 30 min. Absorbance was measured at 734 nm with the help of a BioTek Synergy HT (Winooski, VT, USA) microplate reader. Results were expressed as millimoles of Trolox equivalents per gram of dry extract. Results of the ABTS assay were expressed as Trolox equivalents per gram of extract (mg TE/g extract).

#### 4.7.3. CUPRAC Assay

For the CUPRAC assay, 100 µL of extract/fractions solution were added to the reac- tion mixture [CuCl2 (200 µL, 10 mM), neocuproine (200 µL, 7.5 mM), NH4Ac buffer (200 µL, 1 M, pH 7.0)], and the absorbance was measured at 450 nm after 30 min of incubation at ambient temperature. Furthermore, a blank solution (without the extract/fractions) was prepared and analyzed according to this procedure.

#### 4.7.4. FRAP Assay

For the FRAP assay, 0.05 mL of solution were added to 1 mL of reagent in acetate buffer (0.3 M, pH 3.6), 2,4,6-tris(2-pyridyl)-s-triazine (TPTZ) (10 mM) in 40-mM HCl and ferric chloride (20 mM) with a final concentration at the ratio of 10:1:1 (*v*/*v*/*v*). After incu- bation for 30 min at ambient temperature, the absorbance was measured at 593 nm. Simi- larly, a blank sample (prepared in the same manner but without the extract) was prepared. Furthermore, a blank solution (without the extract/fractions) was prepared and analyzed according to this procedure.

### 4.8. Enzymatic Antioxidants

#### 4.8.1. Catalase Activity (CAT)

The method proposed in Reference [58] was performed to measure the catalase activity. In this method, the hydrogen peroxide (H_2_O_2_) was taken as a substrate and catalase enzymes were used for the decomposition of H_2_O_2_, detected using UV spectrophotometer (UV-1601, Shimadzu, Germany) by calculating the reduction in the absorbance for 5 min at 240 nm. The results of this activity were denoted as µM of consumed H_2_O_2_/min/mg of protein. 

#### 4.8.2. Peroxidase Activity (POD)

For POD activity, the guaiacol was used as a hydrogen. This was performed by measuring the variation at 470 nm for 1 min. The enzymatic activity was described as unit (one activity unit, defined as absorbance at 470 nm, with changes of 0.01 per min) per gram of freshly weighed tissue [59].

#### 4.8.3. Superoxide Dismutase Activity (SOD)

Fresh leaves (1.0 g) were extracted in 50 mM phosphate buffer (pH~7.8) and the homogenate was centrifuged at 15,000× *g* for 10 min. The obtained supernatant was used to assay enzyme activity. The activity of superoxide dismutase was measured according to the method described by [60].

### 4.9. Enzyme Inhibition Assay 

The potential of *P. afra* extract/fractions to inhibit the activity of acetylcholinesterase (AChE), and butrylcholinesterase (BChE), expressed as % inhibition, was determined according to procedures described in the literature, with minor modifications [61]. The detailed experimental methodology is explained below.

#### Acetylcholinesterase and Butrylcholinesterase Inhibition Assay

For the AChE and BChE inhibition assay, after 15 min of incubation at 25 °C, the reaction mixture, comprising 50 µL of the extract/fraction (1mg/mL) solution, 125 µL of DTNB (3 mM), and 25 µL of enzyme solution (0.265 U/mL AChE or 0.026 U/mL BChE) in Tris-HCl buffer (pH 8.0), was incubated at ambient temperature for 15 min. Then, 25 µL of substrate (15 mM acetylthiocholine iodide or butyrylthiocholine chloride) was added to the incubated mixture. The absorbance of the final solution was measured at 405 nm after 15 min. Galanatmine was used as the standard agent for both the acetylcholinesterase and butyryl cholinesterase inhibition assays. Likewise, a blank solution (without the extract/fractions) was prepared and analyzed according to the procedure described by [1].

### 4.10. Thrombolytic Activity

From each human volunteer who had not undergone seven days of anticoagulant and oral contraceptives therapy, venous blood (5 mL) was collected, which was placed in sterile and pre-weighed in six distinct centrifuge tubes. These tubes were incubated for 45 min at 37 °C. After the clot had formed, the fluid from each centrifuge tube was entirely discharged. The clot weight was computed by deducting the weight of empty tube from the tube containing the clot. Streptokinase (SK) was used as a standard to test the thrombolytic activity. Streptokinase (15, 00,000 I.U) was diluted with 5ml sterile water and shaken to mix properly. From this 100 μL, streptokinase (30000 I.U) and distilled water (100 μL) were used as positive and negative controls, respectively, and each sample fraction was separately added to the centrifuge tubes. After that, all tubes were incubated for 90 min at 37 °C and examined for clot lysis. The discharged fluid was discarded after incubation, and the tubes were weighed again to see the weight variation subsequent to clot lysis. Thereafter, the clot lysis percentage was calculated as [62].
Clot lysis percentage = (released clot weight/weight of clot) × 100

### 4.11. Hemolytic Activity

The hemolytic effect of different plant extract fractions was evaluated using the previously described method [58]. Ten milliliters of human blood was collected from human volunteers, added to a sterile, screw-top EDTA tube, and centrifuged at 850 g for 5 min. The upper layer was decanted, and the erythrocytes were rinsed several times with 10 mL cooled isotonic and sterile phosphate-buffered saline (PBS) with a pH of 7.4. The rinsed cells were resuspended in 20 mL sterile and cold PBS. The extracts (1000 µg/1 mL) were added to erythrocytes solution and incubated for 60 min at 37 °C. The absorbance of hemoglobin in the supernatant at 540 nm was used to calculate the hemolysis rate. A total of 0.1% Triton X-100 was used as a positive control and PBS as the negative control. Hemolysis percentage was calculated using the following formula.
Hemolysis percentage = (Ab of sample − Ab of negative control)/Ab of positive control × 100

### 4.12. Gas Chromatography Mass Spectroscopy (GC–MS) Analysis 

The phytoconstituents composition of different *P. afra* extracts was determined using GC MS analysis for Agilent, 6890 series and Hewlett Packard, 5973 mass choosy detection systems, with HP-5MS column (250 m in diameter, 30 m in length 0.25 m in film thickness) was used for compound separation. (70 eV) was used for GC.MS-ionizing radiation of the sample. The temperature at the inlet was 220 °C. The oven temperature steadily increased from 60 °C to 280 °C at a rate of 3 °C/min. The carrier gas was pure helium gas (99.995 percent purity), flowing at a rate of 1 mL/min in constant flow mode. A total of 1.0 L of prepared extracts, diluted with respective solvents, were implanted in a split-less mode at 250 °C; 1.0 L of prepared extracts, diluted with individual solvents, were injected in split-less mode. The compound was discovered using an NIST library search (NIST 14).

### 4.13. Experimental Animals

Wistar albino rats weighing 150–230 g was kept in the animal house of the Pharmacology and Physiology research laboratory at the Islamia University of Bahawalpur’s Faculty of Pharmacy and Alternative Medicine in Pakistan. All the animals used in the study were housed in polycarbonate cages. The typical conditions of temperature (25 °C) and humidity (50–55 percent), as well as exposure to a 12:12-h light and dark cycle, were maintained throughout the study. The animals were fed conventional animal food and given free access to water. To reduce animal stress, animals were acclimatized to the test conditions for one week before the trial began. The institutional research committee approved the study protocols and procedures (PAEC, Pharmacy Animal Ethics Committee).

### 4.14. Anti-Inflammatory Activity

#### 4.14.1. Carrageenan Induced Hind-Paw Edema

Except for the control group, rats received 0.1 mL of newly manufactured, one percent carrageenan, injected into the right hind paw [63]. The paw size of rats was measured using a digital Vernier caliper (Mitutoyo, Japan). The rats were given a vehicle (distilled water 5 mL/kg, i.p), various plant extract doses (25, 50, 100, mg/kg, i.p) and standard indomethacin 15 mg/kg, i.p, half an hour earlier. The thickness of the paws was experimentally validated before the carrageenan injection, at “0 h,” and subsequently at 1, 2, 3, and 4 h. The previously reported formula was used to compute the % inhibition of paw edema [23].
Percentage Inhibition = (Control means − Treated mean)/Control Means × 100

#### 4.14.2. Molecular Docking Procedure

The crystal structures of the Cyclooxygenase 1 protein targets (PDB: 4O1Z, R: 2.40A) and Cyclooxygenase 2 (PDB: 5IKV, R: 2.51 A) were taken from the Protein Data Bank (PDB) (https://www.rcsb.org/, accessed on 25 March 2022) for the docking study. The polar hydrogens were added to the protein. Water, inhibitors, and extra chains were removed from the protein and then converted into the PDBQT format. The ligands were taken from the PubChem (https://pubchem.ncbi.nlm.nih.gov/, accessed on 25 March 2022) to evaluate their potential against the proteins. The ligands were entered into Open Babel and subjected to energy reductions using the PyRx program. After that, the chemicals were converted to PDBQT format for further investigation. Then, the grid box was formed in specific dimensions. Finally, interactions were visualized by the Discovery studio. 

### 4.15. Statistical Analysis

The result values in the tables are represented as mean ± SD (*n* = 6). A two-way ANOVA, followed by Tukey test, was performed using Graph Pad Prism (San Diego, CA, USA) software. All values in the graphs are given in mean ± SE (*n* = 6). *p* < 0.001 (***) and *p* < 0.05 (*) compared to the control. 

## 5. Conclusions

The present study revealed the in vitro antioxidant, thrombolytic, hemolytic activity, and % inhibition of acetylcholinesterase, as well as the butrylcholinesterase and in vivo anti-inflammatory potential of BUTPA extracts. Twenty compounds were identified through GC–MS analysis, which showed many pharmacological activities in in vivo and ex vivo experiments. A high binding affinity was observed for Stigmastan-3, 5-diene, Phthalic acid, 3. Alpha-Hydroxy-5, 16-androstenol. The computed binding energies of the compounds revealed that all the compounds had synergistic effects to prevent inflammation. Therefore, the findings of this study indicated that this plant is an excellent candidate for the treatment of inflammation and pain-related illnesses. The medicinal and pharmacological potential of *P. afra* revealed that it is quite auspicious as a versatile therapeutic plant and should be further investigated.

## Figures and Tables

**Figure 1 molecules-27-02377-f001:**
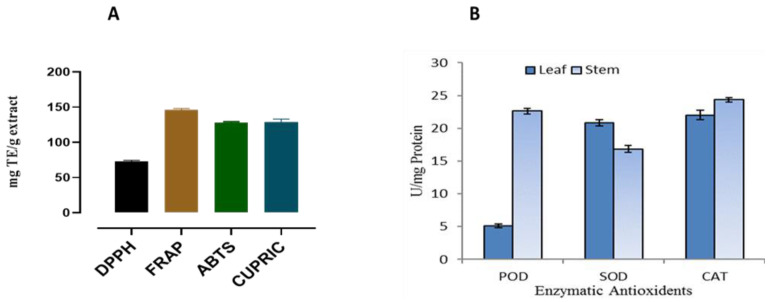
Non-enzymatic (**A**), and enzymatic anti-oxidant (**B**) potential of BUT-PA extracts. For each parameter, vertical bars represent mean data ± SE (*n* = 3).

**Figure 2 molecules-27-02377-f002:**
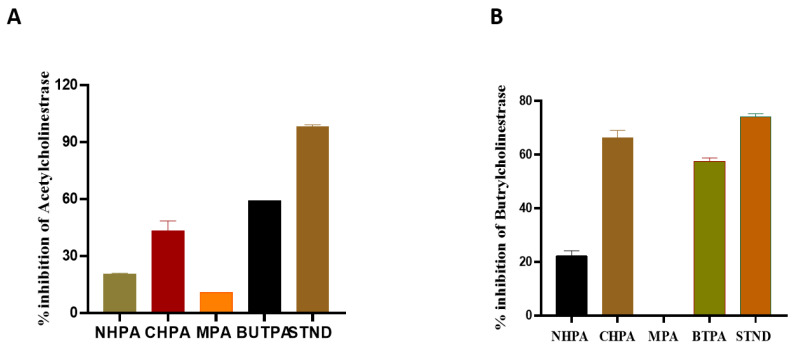
(**A**) Acetylcholinesterase inhibition of galantamine (standard) and extract/fractions, and (**B**) butrylcholinesterase inhibition of galantamine (standard) and extract/fractions of *P. afra*. NHPA, *n*-hexane fraction; CHPA, chloroform fraction; BUTPA, *n*-butanol fraction; STND, galantamine (Standard); MPA, methanolic extract. For each parameter, vertical bars represent mean data ± SE (*n* = 3).

**Figure 3 molecules-27-02377-f003:**
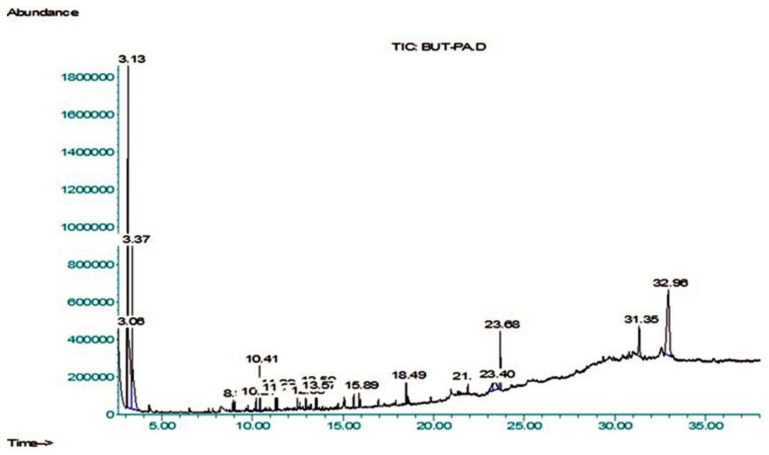
GC-MS chromatogram of BUT-PA extract.

**Figure 4 molecules-27-02377-f004:**
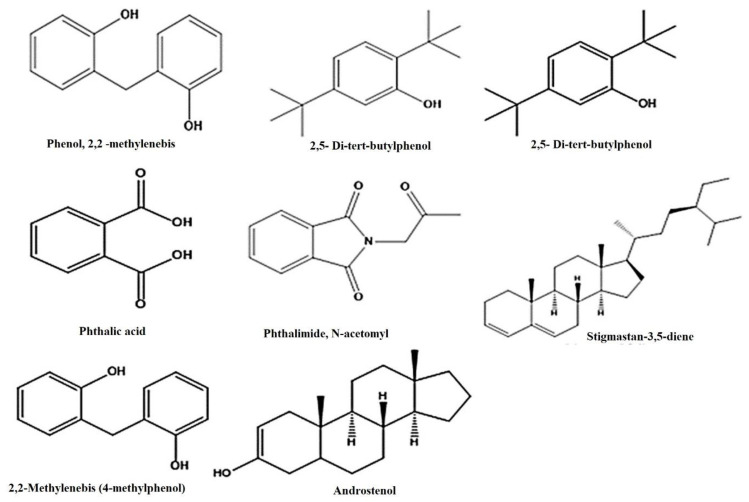
Structures of some phytoconstituents identified in GC-MS of BUT-PA extracts.

**Figure 5 molecules-27-02377-f005:**
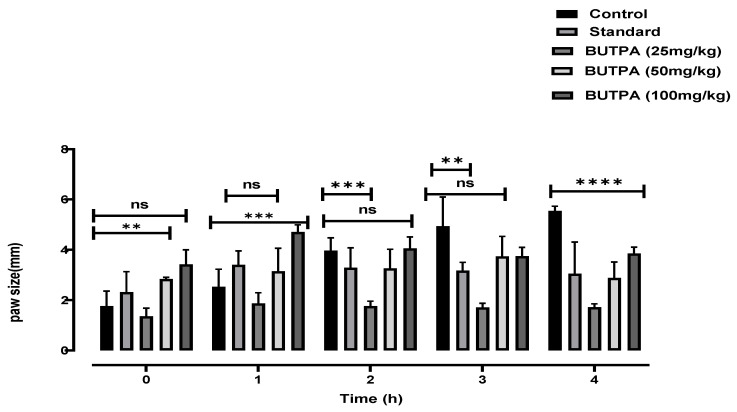
Carrageenan-induced inflammation assay, paw diameter in carrageenan-induced edema was significantly reduced at different doses of plant extract. All result values are represented as mean ± SE (*n* = 6). *p* < 0.001 (*** and ****) and *p* < 0.05 (**) when compared to the control group (two-way ANOVA followed by Tukey test).

**Figure 6 molecules-27-02377-f006:**
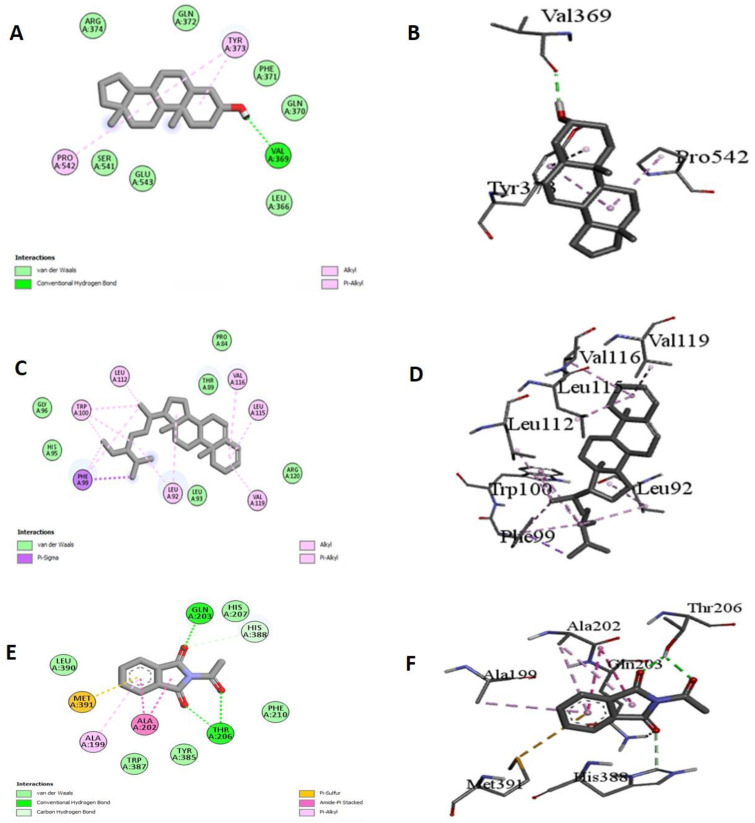
Docking of androstenol, Phthalimide, Stigmastan-3,5-diene performing interaction with Cyclooxygenase-1 (**A**) Androstenol 2d, (**B**) 3d, (**C**) Phthalimide 2d (**D**) (3d) Cyclooxygenase-2 compounds, (3d) Phthalimide2d (**E**), (3d) Cyclooxygenase-2 (**F**) compounds.

**Table 1 molecules-27-02377-t001:** Qualitative phytochemical analysis of different fractions in tissue extracts of *P. afra.*

No.	Metabolites	Test	Extracts			
			MPA	HPA	CHPA	BUTPA
	Primary Metabolites					
1	Carbohydrates	Moloch’s	+	−	+	+
		Fehling’s	+	−	+	+
2	Amino acids	Ninhydrin	−	−	−	−
3	Proteins	Burette	+	+	+	+
4	Lipids	Saponification	+	+	+	+
	**Secondary Metabolites**					
5	Alkaloids	Hager’s	+++	++	++	−
		Wagner’s	+++	++	++	−
		Mayer’s	+++	++	++	−
6	Tannins	Lead Acetate	+++	++	++	−
7	Phenols	Ferric chloride	+++	++	++	+++
8	Flavonoids	Reaction with NaOH	+++	++	++	+++
9	Saponins	Froth	+++	+++	+++	+++
10	Glycosides	Erdmann’s	+++	++	++	−
11	Resins	Acetic Anhydride	++	++	++	++

MPA, methanolic extract; HPA, *n*-hexane fraction; CHPA, chloroform fraction; BUTPA, *n*-butanol fraction; +++, strongly present; ++, moderately present; +, present; −, not present, or not detected.

**Table 2 molecules-27-02377-t002:** Thrombolytic activity of the extract/fractions of BUT-PA extracts and streptokinase blood samples.

No.	Fraction	Thrombolytic Activity (%)
1	HPA	11.21 ± 0.05 B
2	CHPA	7.32 ± 0.15 C
3	MPA	3.41 ± 0.32 E
4	BUTPA	5.51 ± 0.30 D
5	Streptokinase (standard)	91.41 ± 0.06 A

All the procedures were carried out thrice. Data are mean ± standard deviation (*n* = 3). Streptokinase was used as the standard. Significantly different results were exhibited by different letters when compared to the standard (*p* < 0.05).

**Table 3 molecules-27-02377-t003:** Results of Hemolytic Activity of *P. afra.*

No.	Fraction	Hemolytic Activity (%)
1	HPA	5.76 ± 0.15 C
2	CHPA	7.46 ± 0.15 B
3	MPA	9.26 ± 0.15 B
4	BUTPA	8.53 ± 0.25 B
5	Triton × 100	94.53 ± 0.35 A

All the procedures were carried out thrice. Data are mean ± standard deviation (*n* = 3). Triton was utilized as the standard. Significantly different results were exhibited by different letters when compared to the standard (*p* < 0.05).

**Table 4 molecules-27-02377-t004:** GC-MS spectra of GC-MS analysis of BUT-PA extracts (compound identification was based on the NIST library).

Peak Number	RT (min)	Area %	Identified Compounds	Molecular Formula	Molecular Weight	Class	Pharm. Activity
1	3.06	3.93	Ethylbenzene	C_8_H_10_	106.16	Aromatic	*n*/f
2	3.13	33.60	Benzene,1,3-dimethyl-, hexachloro deriv	C_8_H_4_C_l6_	312.8	Aromatic hydrocarbons	*n*/f
3	3.37	16.28	*p*-Xylene	C_8_H_10_	106.16	Benzene Derivatives	CNS depression [23]
4	8.93	0.50	1-Tetradecene	C_14_H_28_	196.37	Alkenes	Antimicrobial [24]
5	10.21	0.68	Pentadecane	C_15_H_32_	212.41	alkane	Antibacterial Antifungal [25]
6	10.41	2.30	2,5-Di-tert-butylphenol 5875-45-6	C_14_H_22_O	206.32	Aromatic Phenols	Antifungal [26]
7	11.29	0.92	7-Hexadecene, (Z)-	C_16_H_32_	224.42	Alkenes	Antimicrobial, antioxidant [27]
8	11.37	0.77	Hexadecane	C_16_H_34_	226.44	Alkanes	Antimicrobial cytotoxic [28]
9	12.48	0.60	Heptadecane	C_17_H_36_	240.5	Alkanes	Antibacterial [1]
10	12.95	0.88	Phthalimide, N-acetonyl	C_10_H_7_NO_3_	189.17	Cyclic imide	Antiinflammatory Analgesic,anti-convulsant [29]
11	13.50	1.09	E-15-Heptadecenal	C_17_H_32_O	252.4	Aldehydes	Antimicrobial [1]
13	15.60	1.09	1,2-Benzenedicarboxylic acid,	C_14_H_22_O_4_Si_2_	310.49	Aromatic Dicarboxylic acid Esters	Antimicrobial [1]
14	15.89	0.89	Dichloroacetic acid	C_19_H_36_C_l2_O_2_	367.4	Fatty Acid Es ters -	Antimicrobial [1]
15	18.49	1.48	Hexadecanoic acid, butyl ester	C_20_H_40_O_2_	312.5	Fatty acid methyl ester	Antioxidant, hypocholesterolemic nematicide [1]
16	21.90	0.93	Phenol,2,2’-methylenebis	C_15_H_16_O_2_	228.29	aromatic organic compound	antimicrobial activity [30]
17	23.39	3.07	9-O-Pivaloyl-N-acetylcolchinol	C_25_H_31_NO_6_	441.5	Aromatic Steroid	Antibacterial Antifungal [31]
18	23.68	4.61	Phthalic acid	C_14_H_16_O_4_	248.27	aromatic dicarboxylic acid,	Antibacterial activity [32]
19	31.35	3.92	Stigmastan-3,5-diene	C_29_H_48_	396.7	tetracyclic triterpene	Anticancer activity [33]
20	32.96	19.95	3.alpha.-Hydroxy-5, 16-androstad	C_19_H_30_O	274.4	endogenous steroid	inhibits CYP2E1 dependent activity [34]

R.T., retention time (minutes); % Area, percent peak area; M.F., molecular formula; M.W., molecular weight; Pharm. Activity, pharmacological activity; Class, chemical class.

**Table 5 molecules-27-02377-t005:** The binding scores and interactions of the examined three compounds, isolated from *P. afra* against COX-1 and COX-2.

Compounds Name	COX-1			COX-2		
	Binding Affinity	Amino Acids	Types of Interaction	Binding Affinity	Amino Acids	Types of Interaction
Stigmastan-3,5-diene	−8.2	LEU A:99 LEU A:92 VAL A:116 VAL A:119 ILE A:89 TRP A:100	Alkyl Alkyl Alkyl Alkyl Alkyl Pi Sigma	−7.3	VAL A:89 ALA A:111 ILE A:112 ILE A:92 LEU A:93	Alkyl Alkyl Alkyl Alkyl Alkyl
Androstenol	−8.1	PRO A:153 CYS A:47 CYS A:41 GLN A:44 ARG A:469 LYS A:468 LEU A:152	Alkyl Alkyl Alkyl H bond Alkyl Alkyl Alkyl	−9.1	CYS A:36 CYS A:47 PRO A:153 ARG A:44	Alkyl Alkyl Alkyl Doner-doner
Phthalimide, N-acetonyl-	−8.1	HIS A: 388 GLN A: 203 ALA A: 202 THR A: 206	Amide pi stacked Van der walls Pi alkyl Pi pi T shaped CH bond	−8.2	HIS A:388 ALA A:202	Pi Pi T shaped Pi alkyl

## Data Availability

Not applicable.

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
