# Peer review of "Phytochemical Profiling, Antioxidant, Anti-Inflammatory, Thrombolytic, Hemolytic Activity In Vitro and In Silico Potential of Portulacaria afra"

_molecules, 2022, doi:10.3390/molecules27082377_

Round 1

Reviewer 1 Report

Dear Author

This is a straightforward study, with the medicinal plant subjects can be seen commonly in the Asian region. The experiments in this study were designed comprehensive. The data was collected and fully statistically processed. 

However, in the discussion, the authors took more time to analyze the obtained data rather than dig into the in vitro biochemical pathways of the extracts.

In short, the article can be accepted as it is in current format, but needs to correct typos.

Best wishes

Author Response

Reviewer#1

This is a straightforward study, with the medicinal plant subjects can be seen commonly in the Asian region. The experiments in this study were designed comprehensive. The data was collected and fully statistically processed. 

Response

Thanks for spending the time on the manuscript and your positive feedback, in the improvement of the manuscript. All thoughtful comments of the respected reviewer have been considered in the revised manuscript.

However, in the discussion, the authors took more time to analyze the obtained data rather than dig into the in vitro biochemical pathways of the extracts.

Response

Thanks for the critical comment. All the suggestions have been incorporated in the discussion section of the revised manuscript.

In short, the article can be accepted as it is in current format, but needs to correct typos.

Response

Thank you so much. All the typographical mistakes have been removed and english language has also improved as per your kind suggestions.

Reviewer 2 Report

Abstract

Latin plant names should be in italics.
Avoid abbreviations or add a list of abbreviations.

Introduction

“didiereaceae” the family name has to be with a Capital letter.

Results

“P. afra” – make it in italic.
The description under Table 1 had shifted. Please correct it.
Please use higher-quality figures.
„The % age acetylcholinesterase inhibition activity of the extract/fractions of P. afra was in the range of (20.55 ± 0.5 to 58.97 ± 0.33).“  and “The % inhibition of BChE of P. afra was calculated as (22.29 ±
1.927 to 57.64 ± 1.192) as shown in (Figure 2).” - The brackets around the results are unnecessary.

Discussion

 “... Consequently, inhibition of free radical inhibition could potentially support the inhibition of carcinogenesis...” - Please, avoid unclear sentences.

Material and methods

The method of SOD activity determination is not correctly described. Reference 56 is not suitable for the presented description.

The manuscript needs thorough English revision and improvements.

Author Response

Reviewer#2

The authors are grateful to you and the reviewers for their valuable comments and suggestions. Their expert opinion has enabled us to strengthen the weaknesses and significantly improve the quality of our manuscript. We are submitting the revised version of our manuscript after modifications. In this version, we strictly observed the comments of reviewers. The details of improvements incorporated in the manuscript according to the comments received are given in the point-by-point responses below. Changes have been highlighted in the revised version of the manuscript. I do hope that the revision will be acceptable for publication in MDPI Molecules and look forward to hearing from you in due course of time.

Abstract

Latin plant names should be in italics. Avoid abbreviations or add a list of abbreviations.

Response

Thanks for the critical comment. Suggestions has been incorporated in revised manuscript.

Introduction

“didiereaceae” the family name has to be with a Capital letter.

Response

Correction has been done.

Results

“P. afra” – make it in italic.

Response

Thank you for suggestions, the name of plant P. afra is italicized in the whole draft.

The description under Table 1 had shifted. Please correct it.

Response

Correction has been done.

Please use higher-quality figures.

Response

High quality images have been added in the revised version of the manuscript.

„The % age acetylcholinesterase inhibition activity of the extract/fractions of P. afra was in the range of (20.55 ± 0.5 to 58.97 ± 0.33).“  and “The % inhibition of BChE of P. afra was calculated as (22.29 ±
1.927 to 57.64 ± 1.192) as shown in (Figure 2).” - The brackets around the results are unnecessary.

Response

Thank you for highlighting the critical mistake, all the unnecessary brackets have been removed in the revised version of the manuscript.

Discussion

 “... Consequently, inhibition of free radical inhibition could potentially support the inhibition of carcinogenesis...” - Please, avoid unclear sentences.

Response

Thank you for highlighting the mistake, correction has been done.

Material and methods

The method of SOD activity determination is not correctly described. Reference 56 is not suitable for the presented description.

Response

The protocol of SOD has been added in the revised version of manuscript.

The manuscript needs thorough English revision and improvements.

Response

All the typographical mistakes have been removed and english language has also improved as per your kind suggestions.